# Response Mechanisms of *Zelkova schneideriana* Leaves to Varying Levels of Calcium Stress

**DOI:** 10.3390/ijms25179293

**Published:** 2024-08-27

**Authors:** Fengxia Yan, Ronghui Jiang, Chao Yang, Yanbing Yang, Zaiqi Luo, Yunli Jiang

**Affiliations:** 1Key Laboratory of National Forestry and Grassland Administration on Biodiversity Conservation in Karst Mountainous Areas of Southwestern China, Guizhou Academy of Forestry, Guiyang 550005, China; yfx19871017@163.com (F.Y.); cyanicjiang@foxmail.com (R.J.);; 2Key Laboratory of Forest Cultivation in Plateau Mountain of Guizhou Province, Institute for Forest Resources and Environment of Guizhou, Guizhou University, Guiyang 550025, China

**Keywords:** *Zelkova schneideriana*, leaf, calcium stress, response mechanisms

## Abstract

Calcium stress can negatively impact plant growth, prompting plants to respond by mitigating this effect. However, the specific mechanisms underlying this response remain unclear. In this study, we used non-targeted metabolomics and transcriptomics to investigate the response mechanisms of *Zelkova schneideriana* leaves under varying degrees of calcium stress. Results revealed that calcium stress led to wilt in young leaves. When calcium stress exceeds the tolerance threshold of the leaf, it results in wilting of mature leaves, rupture of chloroplasts in palisade tissue, and extensive wrinkling and breakage of leaf cells. Transcriptomic analysis indicated that calcium stress inhibited photosynthesis by suppressing the expression of genes related to photosynthetic system II and electron transport. Leaf cells activate phenylpropanoid biosynthesis, flavonoid biosynthesis, and Vitamin B6 metabolism to resist calcium stress. When calcium accumulation gradually surpassed the tolerance threshold of the cells, this results in failure of conventional anti-calcium stress mechanisms, leading to cell death. Furthermore, excessive calcium stress inhibits the expression of CNGC and anti-pathogen genes. The results of the metabolomics study showed that five key metabolites increased in response to calcium stress, which may play an important role in countering calcium stress. This study provides insights into the response of *Z. schneideriana* leaves to different levels of calcium stress, which could provide a theoretical basis for cultivating *Z. schneideriana* in karst areas and enhance our understanding of plant responses to calcium stress.

## 1. Introduction

Calcium ions are essential mineral elements for plants, serving as crucial regulators of plant growth and development as well as a key component of the plant cell wall structure [1]. Calcium acts as an osmotic protective substance in the vacuole, maintaining the stability of the cell membrane and balancing intracellular ions [2,3,4,5]. Furthermore, calcium ions function as important signaling molecules for plant cells to respond to environmental changes [6]. When plants are stimulated by the external environment, there is a specific increase in cytoplasmic calcium ion concentration, which allows cells to transmit information downstream through calcium signals, generating appropriate responses [7].

Despite the crucial role of calcium ions in various physiological processes in plants, high concentrations of calcium ions in the cytoplasm can disrupt the normal physiological functions of plant cells [8]. Calcium stress typically inhibits seed germination, affects plant photosynthesis, reduces plant growth characteristics, and hinders overall plant growth [9,10,11]. Excessive calcium ion content in plant cells can lead to further functional disorders and even cell death [12]. However, these damage mechanisms are complex. At the cellular level, calcium ion stress can disrupt calcium-dependent signaling systems, cause disorders in phosphate-based energy metabolism, and affect the cell skeleton [13,14,15]. Calcium stress causes oxidative damage and compromises cellular membrane stability [16,17]. Furthermore, high levels of calcium stress can affect plant photosynthesis in various ways by reducing cellular chlorophyll levels and disrupting the structure of the photosynthetic membrane [18,19]. Therefore, plants growing in high-calcium environments must possess appropriate physiological mechanisms to avoid excessive calcium absorption [20].

The formation of calcium oxalate plays a crucial role in regulating the concentration of calcium ions in plants. The size and number of calcium oxalate crystals have been identified as important indicators of plant responses to external calcium concentrations in many plants [21,22,23]. Some plants have developed mechanisms to expel calcium oxalate crystals in order to cope with the continuous accumulation of calcium ions. For example, *Pinus massoniana* in karst areas can alternately expel calcium oxalate crystals with pericycle to reduce the concentration of calcium ions [24]. Additionally, under calcium stress, superoxide dismutase (SOD) and catalase (CAT) activities increase to mitigate damage from reactive oxygen species and free radicals in the plant body [25]. High levels of intracellular proline serve as an effective means for plants to maintain osmotic balance under high calcium stress [26]. Molecular biological studies have revealed regulatory mechanisms that enable plants to adapt to high levels of calcium stress. For instance, when chlorophyll content is reduced by calcium stress, *Parachlorella kessleri* maintains a consistent photosynthetic rate by upregulating photosynthetic genes [27]. Furthermore, upregulation of IRT1 expression in *Arabidopsis* may enhance iron ion absorption and improve plant photosynthetic characteristics while enhancing resistance to calcium stress [28]. Plants may also adapt to high levels of calcium stress through the regulation of the cyclic nucleotide gated channel (CNGC) [29].

*Z. schneideriana* is a deciduous broad-leaved tree native to China, which is widely cultivated for its good wood quality and beautiful appearance. [30]. In recent years, *Z. schneideriana* has become a major tree speciescultivated in Guizhou Province, China. Most of Guizhou Province is a karst landform, and the soil parent material is mainly dolomite and limestone, which form soil containing a large amount of calcium ions. The toxic effects of excessive calcium ions pose a challenge to large-scale cultivation of *Z. schneideriana* in karst landform areas. Our previous investigation suggested that, while an excess of calcium ions can hinder the growth of *Z. schneideriana*, some plants are still able to maintain stable survival in karst areas. This indicates that these trees might have developed physiological mechanisms to cope with the toxic effects of calcium ions, although the specific mechanisms involved remain unclear. The mechanism of adaptation to calcium stress in *Z. schneideriana* remains unclear. In this study, the effects of different levels of calcium stress on the leaf tissues of *Z. schneideriana* were studied using tissue sections, and the molecular basis of *Z. schneideriana* in a high-calcium environment was revealed by combining transcriptome and metabolism analyses after calcium stress. The purpose of this study is to provide support for clarifying the adaptation mechanism of *Z. schneideriana* in a karst high-calcium environment and to provide theoretical basis for high-quality planting of *Z. schneideriana* in the karst area, and, at the same time, to provide genetic resources for the creation of a high-calcium tolerant germplasm of *Z. schneideriana*.

## 2. Results

### 2.1. Effects of Different Concentrations of Calcium Stress on the Leaves of Zelkova schneiderian

To investigate the effect of varying concentrations of calcium stress on *Z. schneideriana* seedlings, the seedlings were watered with nutrient solutions containing different concentrations of CaCl_2_ (0, 5, 20, 50, and 100 mmol/L). The seedlings subjected to stress with these solutions are shown in Figure 1A–E. Compared with the control (CK), seedling growth showed no changes after stress with a 5 mmol/L CaCl_2_ nutrient solution. However, when treated with 20 mmol/L CaCl_2_ nutrient solution, slight wilting was observed in the young leaves. Subsequent stress with a 50 mmol/L CaCl_2_ nutrient solution resulted in an increase in both the degree and number of withered young leaves. Notably, stress with 100 mmol/L CaCl_2_ led to the wilting of all leaves. Plants have a tolerance threshold for non-biotic stress, and when the degree of stress is within the tolerance threshold, their growth and development may be inhibited, but they can still survive. However, when the degree of stress exceeds the tolerance threshold, plants gradually die. During the ten-day cultivation period, the CaCl_2_ concentration in the nutrient solution was within the tolerance range of *Z. schneideriana* when it ranged from 0 to 50 mmol/L, and young leaves with low resistance were the first to be affected and withered. However, 100 mmol/L CaCl_2_ exceeded the tolerance threshold, and all leaves gradually died under the influence of calcium stress.

To determine the impact of calcium stress on the plant at the cellular level, we selected mature leaves, stems, and roots from the seedlings and prepared paraffin sections. By observing paraffin sections of leaf, stem, and root tissues, we observed no obvious changes in the roots and stems (Figure 1P–Y), but changes were observed in the leaves (Figure 1F–J). When the concentration of calcium chloride was between 5 and 50 mmol/L, no obvious changes were observed in the leaves (Figure 1F,I,K–N). Interestingly, after cultivation in 100 mmol/L CaCl_2_ nutrient solution, the epidermal cells, palisade tissue, and spongy tissue of the leaves exhibited wrinkling and rupturing, with intracellular chloroplasts also ruptured (Figure 1J,O). Based on these results, we selected mature leaves treated with a control check: 0 mmol/L CaCl_2_ (CK), calcium stress: 20 mmol/L CaCl_2_ (Ca20), and excessive calcium stress: 100 mmol/L CaCl_2_ (Ca100) for further transcriptome and metabolome analysis.

### 2.2. Effects of Different Degrees of Calcium Stress on Gene Transcription Level

We selected leaf samples (three biological replicates) that were consistent with the sections for transcriptome sequencing. Nine cDNA libraries were constructed for sequencing, and 62.21 Gb of clean data were obtained. The clean data for each sample exceeded 6.18 Gb, with a Q30 base percentage > 91.17% (Appendix A). Trinity was used to assemble all clean sample data from scratch, and the assembly results were optimized and evaluated. The number of unigenes detected was 52,688 (Appendix A). Comparing the unigenes in different databases, a total of 30,046 unigenes were annotated. Among them, 17,370 unigenes were annotated in the GO database and 13,393 unigenes in the KEGG database (Figure 2A). The expression levels of different genes were determined using RNA-seq. Most samples exhibited gene expression levels that were relatively close, with a mean value close to log10FPKM = 1 (Figure 2B). Based on the quantitative expression results, DESeq2 was used for intergroup differential gene analysis, with the screening threshold set at |log2FC| ≥ 1 and *p* < 0.05. Subsequently, the differentially expressed genes (DEGs) between different calcium stresses and blank control (CK) were identified. In the Ca20 stress condition, there were 971 DEGs (641 upregulated DEGs and 330 downregulated DEGs), while in the Ca100 calcium stress condition, there were only 258 DEGs (143 upregulated DEGs and 115 downregulated DEGs) (Figure 2C,D).

GO enrichment analysis was conducted to investigate the potential functions of the identified DEGs, which are typically categorized into biological processes, cellular components, and molecular functions. The enriched DEGs under Ca20 stress mainly belonged to molecular function categories in the GO enrichment analysis, such as secondary active transmembrane transporter activity, solute–proton symporter activity, and carbohydrate–cation symporter activity (Figure 3E). However, the enriched DEGs under Ca100 stress were enriched in only two categories: biological processes and molecular functions (Figure 3F). Specifically, they were found to be involved in the phosphorelay signal transduction system, ethylene-activated signaling pathway, and calmodulin binding. In the case of Ca20 stress, 338 DEGs were identified in KEGG enrichment analysis and were associated with 99 metabolic pathways. The most significant pathways included flavonoid biosynthesis, phenylpropanoid biosynthesis, stilbenoid, diarylheptanoid, and gingerol biosynthesis, vitamin B6 metabolism, and brassinosteroid biosynthesis (Figure 3G). Under Ca100 stress, only 87 DEGs were identified in 38 metabolic pathways using KEGG enrichment analysis. These DEGs were found to be mainly involved in plant–pathogen interactions, biosynthesis of various plant secondary metabolites, and phenylpropanoid biosynthesis (Figure 2H).

### 2.3. Calcium Stress Inhibits Photosynthesis in Z. schneideriana

To investigate the impact of calcium stress on photosynthesis at the transcriptomics level, we specifically examined the DEGs involved in the photosynthesis pathway (Figure 3). In this pathway, six DEGs were responsible for encoding five proteins: photosystem II oxygen-evolving enhancer protein 3 (psbQ), photosystem II 10kDa protein (psbR), photosystem II PsbY protein (psbY), photosystem II Psb27 protein (psb27), and plastocyanin (PetE). The expression of these DEGs was downregulated following different levels of calcium stress. Specifically, *psbQ*, *psbR*, *psbY*, and *psb27* are involved in the functioning of photosystem II, whereas *PetE* is involved in electron transfer during photosynthesis. These results indicated that photosynthetic activity was suppressed following exposure to calcium stress.

### 2.4. Calcium Stress Promotes Vitamin B6 Synthesis in Z. schneideriana

Vitamin B6 metabolism is one of the most enriched pathways after Ca20 stress. Meanwhile, vitamin B6, serving as a crucial coenzyme and antioxidant, plays a pivotal role in the growth, development, and response to external stress of plants. Three DEGs were identified in the metabolic pathway of vitamin B6: *phosphoserine aminotransferase* (*serC*), *pyridoxal phosphate phosphatase PHOSPHO2* (*PHOSPHO2*), and *pyridoxal 5′-phosphate synthase pdxS subunit* (*pdxS*) (Figure 4). They were activated under mild calcium stress. Among them, the upregulation of *pdxS* and *PHOSPHO2* may enhance the production rate of pyridoxal 5′-phosphate and its catalytic formation of pyridoxal, while *serC* promoted an alternative pathway for pyridoxal synthesis. Unlike Ca20 stress, the expression levels of these DEGs were not significantly upregulated under Ca100 stress.

### 2.5. Enhanced Flavonoid and Phenylpropanoid Biosynthesis Contribute to Increased Resistance of Z. schneideriana to Mild Calcium Stress

Phenylpropanoid biosynthesis and flavonoid biosynthesis are the two most enriched pathways under calcium stress. Among them, flavonoids play a crucial role in various biological activities within plants and are essential for protecting plants from a range of biological and abiotic stresses such as salinity, drought, ultraviolet light, high temperature, and low temperature. Different levels of calcium stress can lead to the differential expression of 11 genes involved in flavonoid biosynthesis and 14 genes related to phenylpropanoid biosynthesis (Figure 5). During phenylpropanoid biosynthesis, phenylalanine is catalyzed by enzymes such as phenylalanine ammonia-lyase (PAL), cinnamic acid 4-hydroxylase (C4H), and 4-coumarate–CoA ligase (4CL) to form P-Coumaroyl CoA, which serves as a common precursor for both flavonoid and phenylpropanoid syntheses. *PAL*, *C4H*, and *4CL* were upregulated after calcium stress, while shikimate O-hydroxycinnamoyl transferase (HCT), 5-O-(4-coumaroyl)-D-quinate 3′-monooxygenase (*C3′H*), caffeoyl-CoA O-methyltransferase (*CCoAOMT*), and cinnamoyl-CoA reductase (*CCR*) were also upregulated after calcium stress. Interestingly, under Ca20 stress, all DEGs involved in flavonoid biosynthesis, including *CHS*, *chalcone isomerase* (*CHI*), *naringenin 3-dioxygenase* (*F3H*), *flavonoid 3′,5′-hydroxylase* (*F3′5′H*), *flavonoid 3′-monooxygenase* (*F3′H*), *anthocyanidin synthetase* (*ANS*), *flavanone-4-reducta* (*DFR*), *leucoanthocyanidin reductase* (*LAR*), and *anthocyanidin reductase* (*ANR*), were activated. These DEGs may lead to accelerated accumulation of flavonoid substances, thereby promoting increased resistance to calcium stress in *Z. schneideriana* leaves. However, most of these DEGs did not exhibit significant differential expression under Ca100 stress.

### 2.6. Intracellular Ca^2+^ Transport and Accumulation in Z. schneideriana Leaves Is Inhibited under Severe Calcium Stress

Cytosolic Ca^2+^ serves as a regulator of reactive oxygen species (ROS) production and local programmed cell death/hypersensitivity in plants. We attempted to ascertain whether calcium-mediated signaling in the plant–pathogen interaction pathway was affected by calcium stress (Figure 6). Calcium stress induced significant DEGs in 25 genes during plant–pathogen interactions. Under Ca20 stress, one CNGC and EF-hand calcium-binding protein family (*CaMCM*) were upregulated. Additionally, two WRKY transcription factors (WRKY22 and WRKY29), 11 disease resistance proteins RPM1 (RPM1), and one disease resistance protein RPS2 (RPS2) were upregulated under mild calcium stress. Notably, plant–pathogen interactions were identified as the most enriched pathway for DEGs under Ca100 stress, with a total of 20 DEGs enriched compared to control conditions. Unlike mild calcium stress, these genes were predominantly downregulated under severe calcium stress. Specifically, *CNGC*, *CPDK*, *RPM*, and *WRKY33* were downregulated under severe calcium stress. Furthermore, 3-ketoacyl-CoA synthase (*KCS*) was significantly upregulated under Ca100 stress.

### 2.7. Response of Leaf Metabolites to Different Calcium Stresses

To investigate the response of metabolites to different levels of calcium stress, we analyzed the non-targeted metabolic profiles of both CK and calcium-stressed leaf samples. Principal component analysis (PCA) showed that the metabolite measurement results after calcium stress were clearly separated from those of CK, while the results of Ca20 and Ca100 were relatively similar (Figure 7A). Subsequently, differential analysis of the metabolite detection results was conducted. Unlike the transcriptome results, less DEMs were identified between Ca20 and CK (75 significantly upregulated and 85 significantly downregulated) than between Ca100 and CK (87 significantly upregulated and 106 significantly downregulated) (Figure 7C). A Venn diagram illustrated that 53 common DEMs were shared between the two groups (Figure 7D). Changes in these DEMs can better reflect the effective response of *Z. schneideriana* leaves to calcium stress.

Using heat maps to represent the changes in the contents of these 53 DEMs, it is noteworthy that all DEMs exhibited similar trends in accumulation. Specifically, 32 DEMs decreased and 21 DEMs increased under different calcium stresses (Figure 8A). KEGG enrichment analysis revealed that these DEMs were involved in 25 metabolic pathways (Figure 8B). Subsequently, we constructed a KEGG network diagram of these metabolic pathways and observed that the content of DEMs enriched in energy material metabolism (such as starch and sucrose metabolism, amino sugar and nucleotide sugar metabolism, and the citrate cycle (TCA cycle)) were decreased under calcium stress (Figure 8C). Meanwhile, the content of DEMs that enriched in phenylalanine, tyrosine, and tryptophan biosynthesis, ubiquinone and other terpenoid-quinone biosynthesis, tryptophan metabolism, and glycerolipid metabolism increased under calcium stress. This includes L-arogenate, 2-succinyl-5-enolpyruvyl-6-hydroxy-3-cyclohexene-1-carboxylate, cinnavalininate, and phosphatidic acid. Furthermore, DEMs enriched in phenylpropanoid biosynthesis, flavonoid biosynthesis, and glycerophospholipid metabolism were also increased under calcium stress. Specifically, coniferyl alcohol, leucocyanidin, and phosphatidic acid showed an increase in these metabolic pathways. These DEMs may play a crucial role in the leaves of *Z. schneideriana* in response to calcium stress. Notably, phosphatidylglycerol decreased during calcium stress.

### 2.8. Transcription Factor Analysis

Transcription factors play a pivotal role in the regulation of gene expression and are integral to various physiological processes in plants. In this study, we identified 740 transcription factors across 33 families (Figure 9A) to elucidate those involved in the response to calcium stress. Differential expression analysis revealed 37 significantly differentially expressed transcription factors between Ca20 stress and CK (26 upregulated and 11 downregulated), as well as 14 between Ca100 stress and CK (10 upregulated and 4 downregulated), totaling 46 differentially expressed TF, including members of the AP2/ERF (13, 28.3%), MYB (9, 19.6%), NAC (6, 13%), WRKY (5, 10.9%), C2C2 (4, 8.7%), GRAS (2, 4.3%), SBP (2, 4.3%), B3 (1, 2.2%), Bzip (1, 2.2%), GeBP (1, 2.2%), HSF (1, 2.2%), and LOB (1, 2.2%) families (Figure 9B). GO enrichment analysis demonstrated that differentially expressed TFs were notably enriched in the ethylene-activated signaling pathway, regulation of nucleobase-containing compound metabolism, and the phosphorelay signal transduction system (Figure 9C). Additionally, KEGG enrichment analysis showed that only a few TFs were enriched, involving roles in plant–pathogen interactions and the MAPK signaling pathway (Figure 9D). Overall, the majority of AP2/ERF and NAC family transcription factors were upregulated following Ca20 treatment, and a substantial number of AP2/ERF family transcription factors were also upregulated after Ca100 stress, suggesting their potentially crucial roles in mitigating calcium stress.

### 2.9. Quantitative Real-Time PCR (RT-qPCR) Validation

Nine DEGs were selected for RT-qPCR analysis to verify transcriptome data. The results showed that the selected DEGs (*CNGC*, *CPDK*, *KCS*, *DFR*, *LAR*, *PAL*, *PHOSPHO2*, *petE*, and *ptbY*) had basically the same expression patterns in RNA-seq and RT-qPCR, which proved that the transcriptome sequencing results in this study were reliable (Figure 10).

## 3. Discussion

High salinity in plants can lead to secondary stresses, such as ionic stress, osmotic stress, and oxidative stress [31]. Among these, calcium ions play a crucial role as signaling molecules in regulating plant growth and development in response to external stress. Although calcium is generally considered essential for plant growth and development, excessive calcium levels can inhibit normal plant growth and seriously damage photosynthetic characteristics [32]. For example, it was observed that chlorophyll content significantly decreased with increasing CaCl_2_ concentration in *Vicia faba*, *Bacopa monnieri*, and *Ashwagandha* seedlings [2,25,33]. In the present study, we observed a decrease in the number of chloroplasts in the palisade tissues of *Z. schneideriana* leaves after calcium stress. This inhibition of photosynthesis was further evidenced by cleaved chloroplasts, wrinkled cells, and cell breakage following culture in a nutrient solution containing 100 Mm/L CaCl_2_. High osmotic pressure generated inside and outside the cells under salt stress leads to water loss in plants [34]. Cells also initiate adaptive responses, including activation of ion transporters; however, if hypertonic stress exceeds the adaptive threshold of cells, cell death becomes inevitable [35]. In leaves, excessive calcium stress can lead to an influx of calcium that exceeds the cellular tolerance threshold. This disrupts osmotic pressure balance, ultimately resulting in cell atrophy and death [15]. 

As a form of salt stress, calcium stress also exhibits many similarities with other types of salt stress. Previous studies have suggested that salt stress can result in water loss, stomatal closure and reduced photosynthesis [36]. Furthermore, high salt stress can impede the original photochemical reactions and electron transfer processes of PSII, thereby disrupting certain PSII functions [37]. Our study also produced similar findings, with the downregulation of DEGs (*psbQ*, *psbR*, *psbY*, and *psb27*) under calcium stress affecting PSII. This indicates that calcium stress hinders the formation of peripheral protein subunits of PSII, resulting in various functional defects [38]. Additionally, the downregulation of *PetE* affects photosynthetic electron transport, causing a reduction in the photosynthetic rate [39]. In addition to suppressing the synthesis of proteins involved in photosynthesis, we also observed a decrease in phosphatidylglycerol content based on the metabolomic results. PG is a structural lipid found in thylakoid membranes [40], and its deficiency can have further implications for photosynthesis in the chloroplast leaves.

In addition to the photosynthetic system, metal ions in salt stress can disrupt the balance of ROS in plants and induce ROS accumulation, leading to plant damage [41]. For instance, Al, Zn, Cd, Mg, and Ca can stimulate the production of ROS induced by phenoxy radical-mediated lipid peroxidation [42]. Certainly, plants also possess various metabolic pathways that alleviate these detrimental effects. Phenylpropanoid biosynthesis is believed to enhance salt tolerance in *Morus alba* [43]. Salt stress triggers an increase in phenylpropanoid biosynthesis genes, leading to the accumulation of phenylpropanoid metabolites [44]. In flavonoid biosynthesis, *Glycine max* flavonoid compounds increase after exposure to salt stress [45], and upregulation of gene transcription levels regulating flavonoid synthesis has been found to improve the tolerance of *Ginkgo biloba* seedlings to mild saline–alkali stress [46]. In this study, a significant number of DEGs involved in phenylpropanoid biosynthesis (*PAL*, *C4H*, *4CL*, *HCT*, *C3′H*, *CCoAOMT*, and *CCR*) and flavonoid biosynthesis (*CHS*, *CHI*, *F3H*, *F3′5′H*, *F3′H*, *ANS*, *DFR*, *LAR*, and *ANR*) were activated under moderate calcium stress. Meanwhile, the upregulation of these genes led to a significant increase in the content of coniferyl alcohol in phenylpropanoids biosynthesis and leucocyanidin in the flavonoid biosynthesis. They are phenolic compounds that transfer electrons to peroxidase to clear H_2_O_2_ [42]. Leucocyanidin, an important flavonoid compound, has been shown to increase in red rice following exposure to salt stress [47]. Flavonoids not only act as regulators of ROS homeostasis, but also serve as signaling molecules that activate defense-related signaling pathways and regulatory mechanisms, thereby enhancing plant tolerance and resistance against adverse stresses [48,49]. Furthermore, coniferyl alcohol has been demonstrated to mediate plant responses to abiotic stresses, such as salt and alkali, when applied exogenously [50], while also inducing alternative oxidase activity in *Arabidopsis*, leading to reduced ROS levels [51]. 

Furthermore, all DEGs involved in vitamin B6 metabolism (including *serC*, *PHOSPHO2*, and *pdxS*) were upregulated under mild calcium stress. In *Arabidopsis thaliana*, vitamin B6 is thought to regulate intracellular and extracellular ion homeostasis by controlling the activity of ion transporters [52]. Previous studies have shown that the VB6 content increases under salt stress in *Zea mays* roots and is involved in regulating the metabolic levels of abscisic acid (ABA) and ROS under salt stress [53]. Under mild calcium stress conditions, *Z. schneideriana* leaves triggered the synthesis of VB6, which helped to maintain plant ROS and ion homeostasis. 

Calcium ions function as specialized signaling molecules that activate the plant immune system through a specific pathway when the plant is exposed to other stresses [54]. When attacked by external pathogens, *CNGC* genes are upregulated, leading to a specific increase in cytoplasmic calcium ions, triggering the plant immune response to pathogens (PTI) [55]. In fact, this study revealed a distinctive phenomenon of immune gene activation triggered by calcium stress, which seems to be a characteristic of calcium ion stress that distinguishes it from other salt stresses. Our findings suggest that mild calcium stress also induces the upregulation of *CNGC*. At the same time, the genes involved in calcium-mediated immune response were also activated, including *CaMCM*, WRKY22, WRKY29, *RPM*, and *RPS*. Among these genes, *RPM* plays a key role in regulating the plant immune response to pathogens by promoting the accumulation of calcium ions in the cytoplasm [56]. Additionally, WRKY serves as an important transcription factor involved in plant transcriptional regulation and pathogen defense [57]. Researchers have found that higher concentrations of exogenous calcium in *Arabidopsis* lead to increased expression of defense-related genes [58]. This calcium stress-induced immune response is likely to result in the accumulation of ROS, which could damage the leaves of the *Z. schneideriana*.

Notably, the number of DEGs identified under excessive calcium stress was significantly lower than that under moderate calcium stress. Slice results indicate that excessive calcium stress surpasses the cellular tolerance threshold of the plant, leading to a gradual failure of the regulatory mechanisms maintained under moderate calcium stress, resulting in a milder transcriptomic response. However, the plant–pathogen interaction pathway remains highly enriched under excessive calcium stress in the leaves of *Z. schneideriana*. Within this pathway, the CNGC functions as an inward calcium channel that prevents calcium accumulation in leaves [59]. The downregulation of *CNGC* suggests inhibition of calcium ion inflow into the cytoplasm. Additionally, *CPDK*, *CaMCM*, WRKY33, and *RPM* were downregulated under excessive calcium stress, indicating the inhibition of the pathogen defense mechanism mediated by calcium signaling. Furthermore, *KCS* is upregulated under excessive calcium stress conditions. Previous studies have suggested its involvement in the response to salt stress [60,61], as well as in promoting chloroplast matrix formation in *Corchorus capsularis* [62]. Although *KCS* may play a role in maintaining leaf stability under calcium stress, further research is needed to elucidate its specific functions.

Salt stress induces metabolic disorders in plants, leading to damage of the redox system. This results in the accumulation of reactive oxygen species and other substances, which in turn affect photosynthesis [63]. Plants respond to salt stress by increasing the accumulation of metabolites and exudation of organic acids as part of their defense mechanisms [64,65]. Our study found that although the transcriptome response to moderate calcium stress was more excessive than that to excess calcium stress, the metabolome results were the opposite. More DEMs were identified in leaves subjected to excess calcium stress than in those subjected to moderate calcium stress, indicating that the leaves in the latter were able to resist calcium stress before wilting. Analysis of the KEGG metabolic pathway network revealed a decrease in the accumulation of shared DEMs within energy metabolism-related pathways under calcium stress. This decrease may be attributed to suppressed photosynthesis in leaves, leading to diminished synthesis of energy materials. Among the significantly increased shared DEMs, in addition to the ones mentioned earlier, such as coniferyl alcohol and leucocyanidin, there were L-arogenate, 2-succinyl-5-enolpyruvyl-6-hydroxy-3-cyclohexene-1-carboxylate, cinnamavalininate, and phosphatidic acid. Although some studies have revealed that the content of L-arogenate and cinnavalininate changes significantly in organisms under stress, their functions remain unclear [66,67,68]. It is worth noting that phosphatidic acid is a membrane phospholipid that rapidly increases in response to salt stress and plays a role in regulating ion efflux and reactive oxygen species production by interacting with target proteins [69].

A substantial body of research has demonstrated the widespread involvement of AP2/ERF and NAC transcription factors in plant tolerance responses to various abiotic stresses such as salt, drought, high temperature, and cold [70,71,72,73,74]. Among these, AP2/ERF is commonly recognized as a positive regulator of transcriptional activity. Studies have indicated its role in ethylene-mediated salt tolerance, with overexpression of ERF being beneficial for enhancing plant resilience to salt stress [75,76,77,78]. The response of NAC to salt stress has been observed in several studies. Overexpression of NAC can lead to the accumulation of stress-protective substances (e.g., proline and betaine), enhancement of the antioxidant system, promotion of ethylene synthesis, and increased salt stress tolerance [79,80,81]. In this study, multiple AP2/ERF and NAC transcription factors were found to respond to salt stress. They were activated following calcium stress exposure, suggesting their participation in conferring tolerance to *Z. schneideriana* leaves under calcium stress conditions by mitigating damage caused by calcium ion accumulation in the leaves.

## 4. Materials and Methods

### 4.1. Plant Materials

In mid-November 2022, *Z. schneideriana* seeds were harvested in Duyun City, Guizhou Province, southwest China (26°15′34″ N, 107°31′07″ E). In May 2023, the seeds were washed with water and sown in nutrient soil (the primary components of this product are peat moss, fertilizer, and vermiculite, resulting in a loose texture and a pH range of 6.2 to 7.0). After 3 months of cultivation, 25 seedlings with similar growth status and a height of 10 cm were selected as experimental samples and divided into 5 groups, with 5 seedlings in each group. All *Z. schneideriana* seedlings were watered with a half-strength Hogland nutrient solution at a rate of 300 mL per pot. After 15 days of continuous watering, a 1/2 strength Hortogran nutrient solution was used to prepare nutrient solutions with varying concentrations of calcium chloride (0, 5, 20, 50, and 100 mmol/L), which were administered daily at 5 PM, with each group receiving a 300 mL application of the nutrient solution. This study was carried out in a transparent greenhouse with consistent temperature, humidity, and lighting conditions. An automated irrigation system was implemented to ensure even watering of the plants, thereby maintaining consistent water conditions for the seedlings. After 10 days of watering, 3 seedlings from each group were selected to pick mature leaves for a total of nine samples in each group; three samples were preserved in FAA fixation solution for more than three days to make slices; three samples were frozen in liquid nitrogen and stored in an ultra-low temperature refrigerator (−80 °C) for RNA extraction; and three samples were directly stored in an ultra-low temperature refrigerator (−80 °C) after picking for determination of the non-targeted metabolic group.

### 4.2. Preparation and Observation of Plant Slices 

The leaves were removed from the FAA fixative, cut into segments with a blade, and dehydrated step-by-step with 70%, 85%, 95%, and 100% ethanol concentration gradients. They were then made transparent with xylene, soaked, and embedded in paraffin. The leaves were sliced (8 μm thick) using a Leica RM2235 paraffin slicer (Leica, Wetzlar, Germany). The wax strips were unrolled at 47 °C and dried at 40 °C. Subsequently, the slices were stained with saffron O (China) and fast Green (China), sealed with neutral gum, observed under a Leica DM2500 microscope (Leica, Wetzlar, Germany), and photographed. Based on the slice results, samples (0 mmol/L CaCl_2_, 20 mmol/L CaCl_2_, and 100 mmol/L CaCl_2_) were selected for subsequent metabolomics determination and RNA extraction.

### 4.3. Extraction and Determination of Metabolites

A 50 mg sample was added to 1 mL of extraction solution (methanol, acetonitrile, and water in a volume ratio of 2:2:1, with an internal standard concentration of 20 mg/L) and mixed for 30 s. The mixture was ground using an Wonbio-800 abrasive grinder (Wonbio, Shanghai, China) at 45 Hz for 10 min. Subsequently, the mixture was left to stand at −20 °C for 1 h and centrifuged for 15 min at 4 °C and a speed of 12,000 r/min using a PRO1905R centrifuge (Everfuge, Shanghai, China). Following this step, 500 μL of the supernatant was transferred to an EP tube; the liquid in the EP tube was dried and then reconstituted with 160 μL of extract (acetonitrile–water volume ratio of 1:1). The solution was subjected to ultrasonication for 10 min before being centrifuged again under the previous settings. Finally, 120 μL of the supernatant was transferred to a sample bottle; 10 μL of liquid from different samples was mixed together to prepare Quality Control (QC) samples for testing purposes. Finally, mass spectrometry data were obtained using a high-resolution mass spectrometer (Xevo G2-XS QTOF, Waters, Eschborn, Germany).

### 4.4. Identification, Annotation, and Analysis of Metabolites

Raw data were collected using Mass Lynx V4.2 (Waters, Milford, MA, USA), while Progenesis QI3.0 (Waters, Milford, MA, USA) and the Biomark self-built library (Biomarker, Beijing, China) were used to extract peaks, compare compounds, and identify them. Principal component analysis (PCA) was used to evaluate the producibility of different samples. Based on the grouping information, differential multiples (FC) values were calculated, and the significance *p*-values of different compounds were calculated using a *t* test. OPLS-DA modeling was performed using the R language package ropls, and VIP values were calculated through multiple cross-validations. The selection criteria for the differential metabolites were FC > 1.0, *p*-value < 0.05, and VIP > 1. The hypergeometric distribution test was used to evaluate metabolites that were associated with KEGG pathway enrichment.

### 4.5. RNA Extraction and Transcriptome Sequencing

RNA was extracted from the samples using the RNAprep Pure Plant Kit (Tiangen, Beijing, China) according to the manufacturer’s instructions. The quality, quantity, and integrity of the RNA were assessed using a NanoDrop 2000 spectrophotometer (Thermo, Waltham, MA, USA) and an Agilent 2100/LabChip GX (Agilent, Santa Clara, CA, USA) to ensure that they met the sequencing standards. Subsequently, cDNA libraries were constructed using the Illumina HiSeq 2000 platform and were sequenced. Clean reads were obtained by removing spliced and low-quality sequences from the libraries. Trinity [82] was used to assemble these clean reads into unigenes for further analysis.

### 4.6. Gene Function Annotation and Differential Expression Analysis

The unigenes were annotated via the Kyoto Encyclopedia of Genes and Genomes (KEGG) and Gene Ontology (GO) using DIAMOND [82]. Sequencing reads were aligned to the unigene library, and their expression levels were estimated using Bowtie [83] and RSEM [84]. The expression abundance of each corresponding unigene was quantified using FPKM values. Differential expression analysis between different groups was performed using the DESeq R package version 1.10.1 [85], with significant differential expression determined by *q* < 0.05 and |log2FoldChange| > 1 criteria. The GO seq R package version 1.10.0 and KOBAS version 2.0.12 [86,87] were utilized for GO enrichment analysis and KEGG pathway enrichment analysis of DEGs.

### 4.7. Validation of Transcriptome Results via RT-qPCR

Nine DEGs were selected for the validation of their expression levels using RT-qPCR to ensure the reliability of the RNA-seq findings. Primers were designed using Primer Premier 5.0 software (PREMIER, Palo Alto, CA, USA) (Appendix A). The experiment was conducted using the Talent qPCR PreMix (SYBR Green) kit (Tiangen, Beijing, China) and the CFX96 Touch Real-Time PCR System (Bio-Rad, Hercules, CA, USA), following the manufacturer’s instructions. The transcription level of each gene was calculated using UBC as an internal control.

### 4.8. Data Visualization and Image Editing

TBtools-II [87] was utilized to generate a heatmap and concurrently normalize the data during graph creation. Origin 2022 (OriginLab, Northampton, MA, USA) and Adobe Photoshop 2023 (Adobe, San Jose, CA, USA) were used to complete the remaining image creation and layout.

## 5. Conclusions

In this study, we investigated the response mechanisms of *Z. schneideriana* leaves to calcium stress (Figure 11). Our findings revealed that young leaves wilt under calcium stress, and the severity of wilt in young leaves increases with increasing calcium concentrations. Under excessive levels of calcium stress, mature leaves begin to wilt, leading to ruptured chloroplasts in the palisade tissue, extensive cell wrinkling, and death. Transcriptomic analysis indicated downregulation of the transcription levels of PS II and electron conduction genes under calcium stress. Furthermore, some genes related to pathogen defense were also upregulated. Leaves induce gene upregulation of phenylpropanoid biosynthesis, flavonoid biosynthesis, and vitamin B6 metabolism to enhance their resistance to calcium stress. However, these mechanisms fail under conditions of excessive calcium stress. Excessive calcium stress inhibited the expression of CNGC and pathogen defense genes, which may help alleviate cell death caused by calcium stress. Non-targeted metabolomics analysis revealed an increase in five key metabolites under calcium stress, with coniferyl alcohol and leucocyanidin potentially exerting a significant influence on enhancing plant resistance to calcium stress.

## Figures and Tables

**Figure 1 ijms-25-09293-f001:**
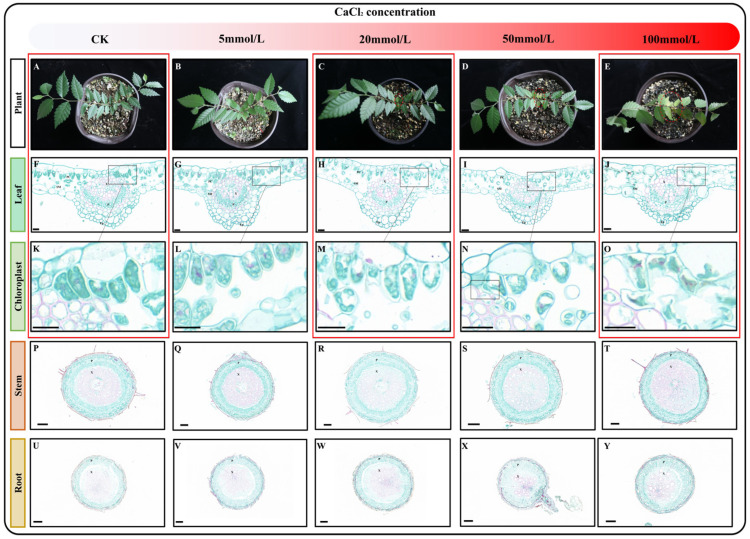
Characteristics of seedlings after exposure to varying levels of calcium stress. (**A**–**E**) Growth status of seedlings cultivated in nutrient solutions containing varying concentrations of CaCl_2_. (**F**–**J**) Paraffin sections of leaves following exposure to different levels of calcium stress. (**K**–**O**) Effects of varying levels of calcium stress on chloroplasts. (**P**–**T**) Paraffin sections of stems following exposure to different levels of calcium stress. (**U**–**Y**) Paraffin sections of roots following exposure to different levels of calcium stress. Note: P—phloem, X—xylem, Ep—epidermis, PC—parcel, and SM—sponge. The scale of F-O: 20 μm; the scale of P-Y: 100 μm. Red boxes indicate samples designated for transcriptome sequencing.

**Figure 2 ijms-25-09293-f002:**
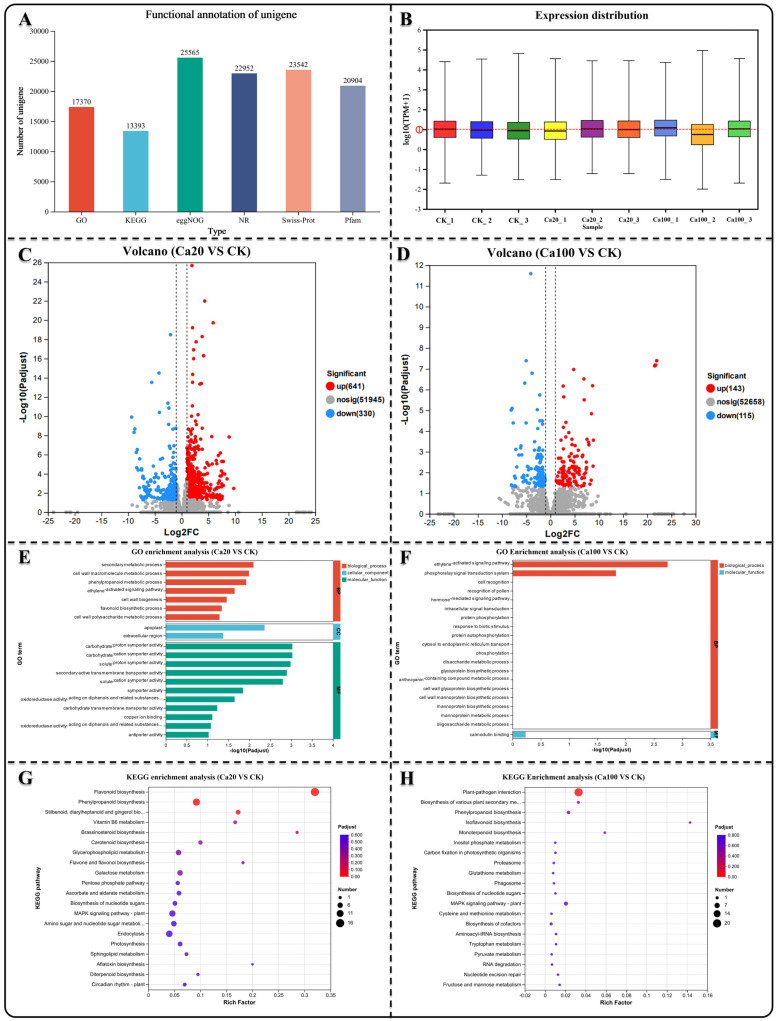
Gene difference analysis, GO enrichment analysis, and KEGG enrichment analysis. (**A**) Quantity of unigenes annotated across various databases. (**B**) Distribution of unigene expression levels in diverse samples. (**C**,**D**) Identification of DEGs in different groups. (**E**,**F**) GO enrichment analysis of DEGs in different contrasts showing the top 20 enrichment results. (**G**,**H**) KEGG enrichment analysis of DEGs in different groups showing the top 20 enrichment results.

**Figure 3 ijms-25-09293-f003:**
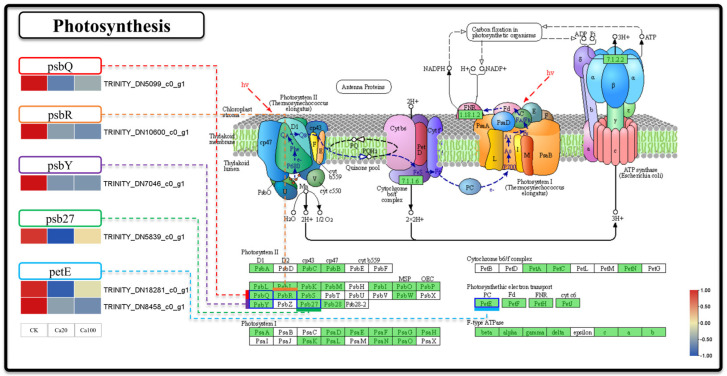
Photosynthesis. Note: the heatmaps from left to right represent CK, Ca20, and Ca100, respectively. In these representations, red indicates upregulation, and blue indicates downregulation.

**Figure 4 ijms-25-09293-f004:**
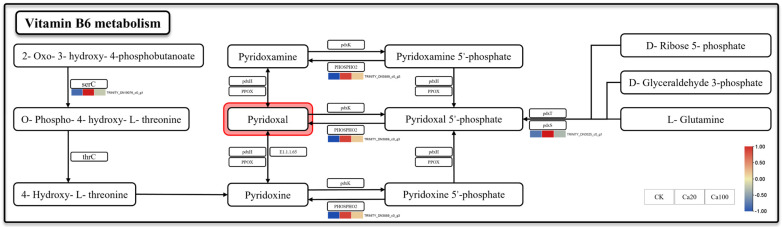
Vitamin B6 metabolism. Note: the heatmaps from left to right represent CK, Ca20, and Ca100, respectively. In these representations, red indicates upregulation, and blue indicates downregulation.

**Figure 5 ijms-25-09293-f005:**
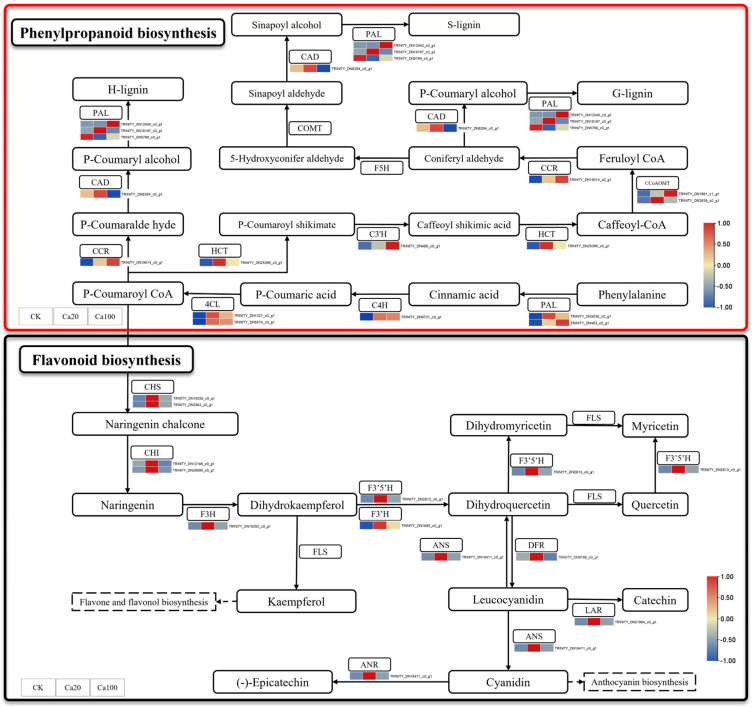
Phenylpropanoid biosynthesis and flavonoid biosynthesis pathway. Note: the heatmaps from left to right represent CK, Ca20, and Ca100, respectively. In these representations, red indicates upregulation, and blue indicates downregulation.

**Figure 6 ijms-25-09293-f006:**
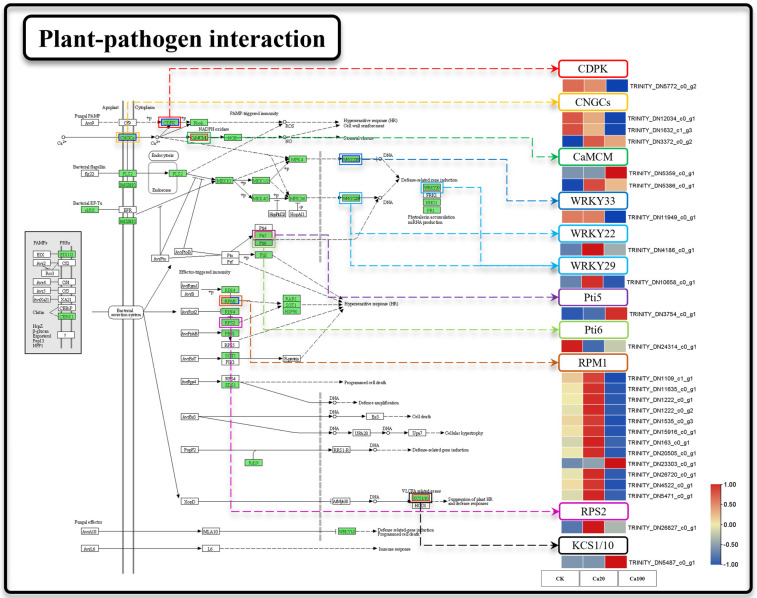
Plant–pathogen interaction. Note: the heatmaps from left to right represent CK, Ca20, and Ca100, respectively. In these representations, red indicates upregulation, and blue indicates downregulation.

**Figure 7 ijms-25-09293-f007:**
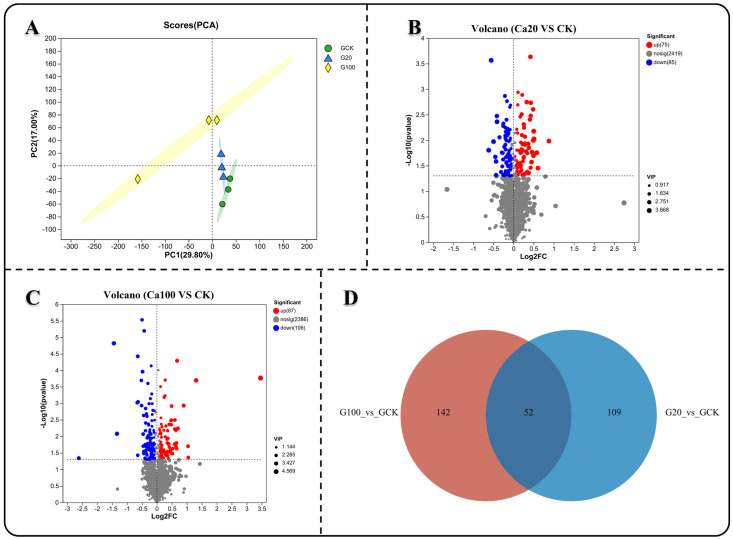
Metabolite differences analysis under varying levels of calcium stress. (**A**) Principal component analysis (PCA) of samples subjected to different levels of calcium stress. (**B**,**C**) Metabolite difference analysis between mild and severe calcium stress groups compared with the blank control. (**D**) Venn diagram illustrating common genes among the different groups.

**Figure 8 ijms-25-09293-f008:**
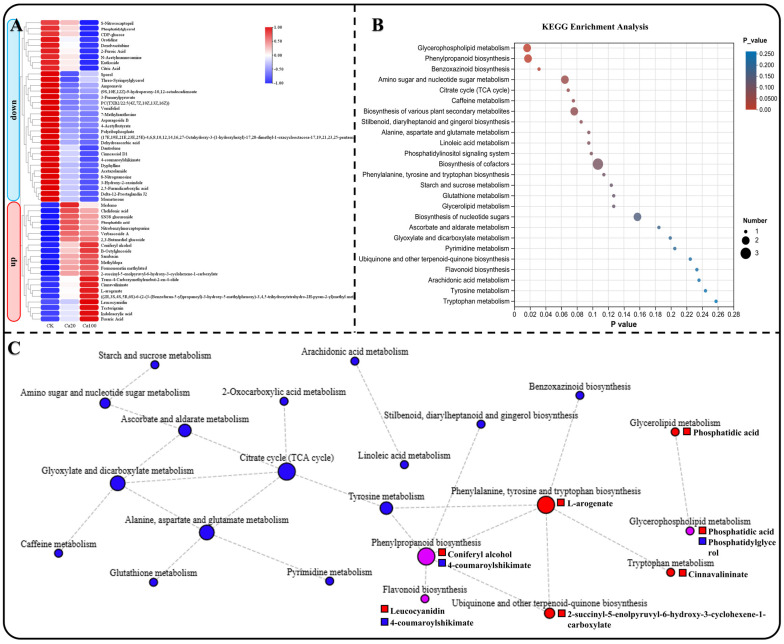
Enrichment analysis of common DEMs. (**A**) Heatmap depicting the expression of common genes enriched in KEGG. (**B**) KEGG enrichment analysis of common DEMs among different groups, presenting the top 20 enrichment results. (**C**) KEGG network diagram of common DEGs, with circles denoting metabolic pathways and squares representing metabolites. Red, blue, and purple indicate upregulation, downregulation, and both upregulation and downregulation, respectively.

**Figure 9 ijms-25-09293-f009:**
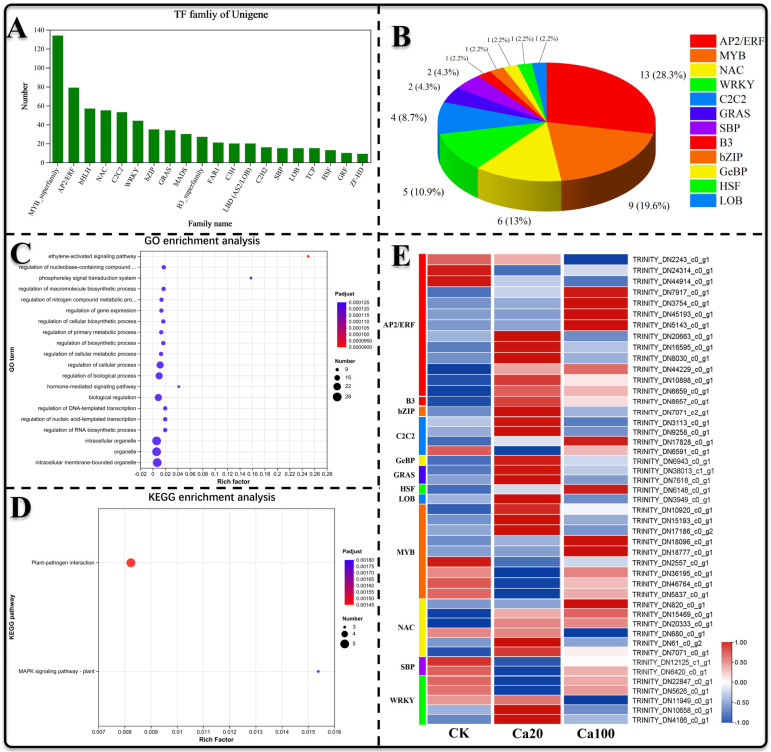
Transcription factor analysis. (**A**) Analysis of the distribution of transcription factors across different families; (**B**) number and percentage of differentially expressed transcription factors within each family; (**C**) GO enrichment analysis of the identified differential transcription factors; (**D**) KEGG enrichment analysis of the identified differential transcription factors; (**E**) expression patterns of the differentially expressed transcription factors.

**Figure 10 ijms-25-09293-f010:**
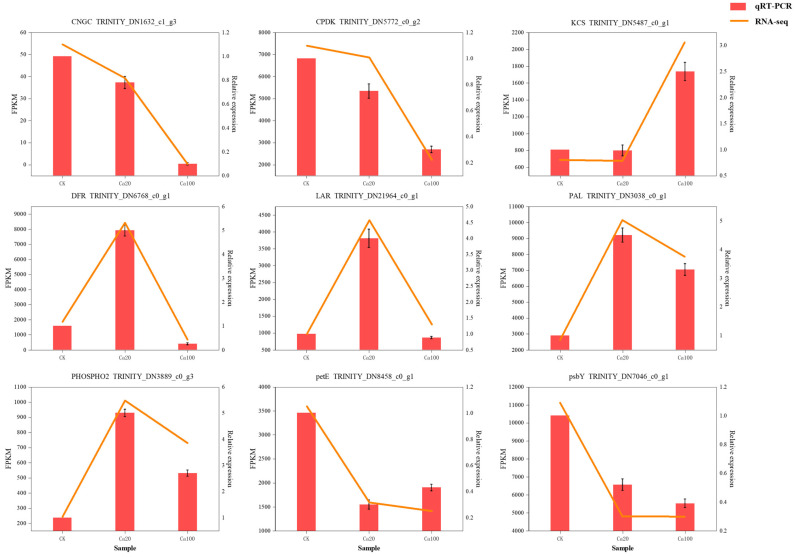
qRT-PCR validation of nine DEGs.

**Figure 11 ijms-25-09293-f011:**
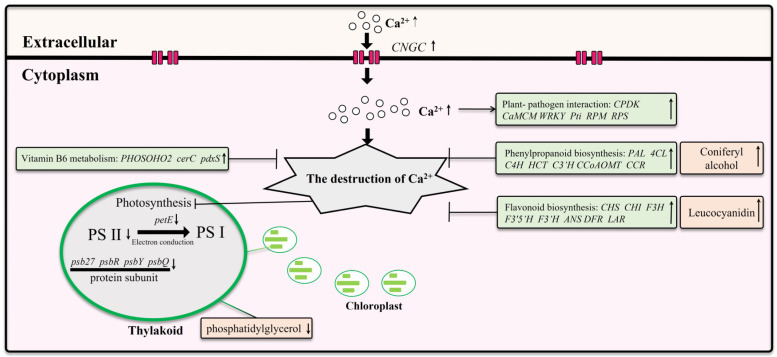
The response of *Z. schneideriana* leaf to calcium stress.

## Data Availability

The raw RNA-seq datasets generated in this study can be found at NCBI SRA under the project number: PRJNA1095693. [https://www.ncbi.nlm.nih.gov/sra/PRJNA1095693; accessed on 22 April 2024].

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
