# Peer review of "Response Mechanisms of Zelkova schneideriana Leaves to Varying Levels of Calcium Stress"

_ijms, 2024, doi:10.3390/ijms25179293_

Round 1

Reviewer 1 Report

Comments and Suggestions for Authors

The paper investigates the physiological and molecular responses of Zelkova schneideriana leaves to different concentrations of calcium chloride (CaCl2). Using non-targeted metabolomics and transcriptomics, the study reveals that calcium stress leads to wilting and cellular damage in leaves, with severe stress causing cell death. The authors report the downregulation of photosynthesis-related genes and the upregulation of genes involved in phenylpropanoid biosynthesis, flavonoid biosynthesis, and Vitamin B6 metabolism under mild stress. I believe that the paper presents a significant contribution to understanding Z. schneideriana's responses to calcium stress, with potential applications in forestry. Before its publication, the authors are requested to consider the following points:

1. Calcium-specific Effects

The authors aim to provide insights into the adaptation mechanisms of Z. schneideriana in high-calcium environments and support its cultivation in karst areas, which is important for local forestry, adding to its unique contribution. A major weakness of this study is the lack of focus on the specific impacts of calcium on the plants. Calcium, as a divalent cation, plays unique and important roles in cells, such as in signaling mechanisms at low concentrations and in the structural integrity of the cell wall at high concentrations. It is unfortunate that physiological experiments were conducted only with calcium. To distinguish calcium-specific effects from non-specific ones, such as osmotic or ionic strength stress, it would be beneficial to compare other divalent cations such as magnesium (Mg) or other similar ions.

2. Phosphate Availability in Soil at High Calcium Concentrations

In general, plants growing in high-calcium soils, where calcium phosphate precipitates and potentially leads to phosphate deficiency, have developed several adaptive strategies to cope with this challenging condition. Tolerant plants can manage phosphate deficiency in high-calcium soils through multiple mechanisms: mycorrhizal associations, release of organic acids (https://doi.org/10.1016/S0168-9452(00)00347-2), acidifying the rhizosphere by releasing protons, and utilization of phosphate-solubilizing microorganisms (http://www.springerplus.com/content/2/1/587). This manuscript does not mention such indirect impacts of high-calcium soils but only states and discusses direct effects. It would be beneficial if the authors included indirect impacts, such as a decrease in soil phosphate availability, in their scope.

3. Calcium-induced Oxidative Stress

Calcium is necessary for the plant cell wall through binding to polysaccharides to enhance structural integrity. Polyphenols contribute to this rigidity of cell wall structures through lignification, which requires radical chain reactions. However, divalent cations may induce oxidative stress by increasing the lifetime of phenol radicals, as previously shown (https://doi.org/10.1016/S0300-483X(02)00196-8). Since one of the major findings of this study includes enhanced phenylpropanoid or flavonoid biosynthesis, it would be beneficial to discuss the possible contribution of polyphenols or flavonoids in calcium responses.

Reviewer 2 Report

Comments and Suggestions for Authors

Overall, the work is well written. I pay attention to improving the quality and readability of the figures. The font size and in some places the resolution is too low. In addition, the work contains many typos - this must be corrected.

- I propose incorporating hypothesis testing before stating the aim of the paper. This approach would enhance the logical flow of the manuscript, providing a clearer context for the subsequent research objectives.

- In the methodological part, it is necessary to complete the data on the conditions of cultivation in particular - whether the conditions (photoperiod, temperature, humidity) were controlled and to what extent.

Reviewer 3 Report

Comments and Suggestions for Authors

1) L 21-22”” When calcium accumulation gradually surpassed the tolerance threshold of the cells.” – unfinished thought?

2) Improve readability Figures 2, 7-10.

3) L 204: Adapt the literature citation in the text of the manuscript to the requirements of the journal.

4) I suggest the Authors consider rewriting the Discussion section. The Authors confront the results of their own research on the effects of calcium stress mainly with the effects caused by salt stress, which in my opinion is not appropriate.

5) L 447: „the seeds were (…) sown in nutrient soil” – please provide the physicochemical properties of this soil.

6) The study included 5 levels of calcium chloride concentrations (0, 5, 20, 50, and 100 mmol/L) (L 91, L 453), but the results presented in the manuscript only concern the effect of concentrations: 0 (CK), 20 (Ca20), and 100 mmol/L (Ca100). Since "seedling growth showed no changes after stress with a 5 mmol/L CaCl2 nutrient solution" compared with the control (CK) (L 92-93), the authors could have omitted this concentration. However, please explain why the results for the concentration of 50 mmol/L were not shown in the publication.

7) Supplementary Materials: include explanations of the abbreviations used in the tables below the tables.

Round 2

Reviewer 1 Report

Comments and Suggestions for Authors

After incorporating the comments and suggestions of the reviewers, the authors have revised the manuscript, resulting in significant improvements. The overall revision is satisfactory.